

# Supplementation of PUFA extracted from microalgae for the development of chicken patties

Sidra Dr[1], Muhammad Muneeb Zaman[2], Zunaira Farooq[2], Amina Hafeez[3], Muhammad Wasim Sajid[4], Muhammad Rizwan Tariq[5], Shinawar Waseem Ali[5], Sajid Ali[6], Muhammad Shafiq[7], Madiha Iftikhar[8], Waseem Safdar[9], Umair Ali[10], Maria Kanwal[5], Zujaja Umer[5] and Zunaira Basharat[5]

[1] Shalamar Medical and Dental College, Lahore, Pakistan
[2] Sahiwal Medical College, Sahiwal, Pakistan
[3] Services Institute of Medical Sciences, Lahore, Pakistan
[4] Biosciences, COMSATS University Islamabad, Sahiwal, Punjab, Pakistan
[5] Department of Food Sciences, University of the Punjab, Lahore, Pakistan
[6] Department of Agronomy, University of the Punjab, Lahore, Pakistan
[7] Department of Horticulture, University of the Punjab, Lahore, Pakistan
[8] Department of Diet and Nutritional Sciences, Ibadat International University, Islambad, Pakistan
[9] Department of Biological Sciences, National University of Medical Sciences, Rawalpindi, Pakistan
[10] Dept of Food Science and Technology, Faculty of Agriculture and Environment, Islamia University, Bahawalpur, Pakistan

Corresponding authors
Muhammad Wasim Sajid,
muhammad.wasim@cuisahiwal.edu.pk
Muhammad Rizwan Tariq,
rizwan.foodsciences@pu.edu.pk

## ABSTRACT

In recent years, there has been a growing interest in development of a diverse range of foods that are enriched with omega-3 fatty acids. It is widely recognized that through dietary interventions, the lipid fraction of food can be modified to enhance its nutritional content. This study is aimed to develop chicken patties enriched with poly unstaurated fatty acids (PUFAs) extracted from microalgae aurintricarboxylic acid (ATA) concentration of 0% (T0), 1% (T1), 2% (T2), and 3% (T3). All treatments were stored at $-18$ °C for one month and analysed at an interval of 0, 10, 20, and 30 days to assess the effect of PUFAs supplementation on physicochemical, oxidative, microbiological and organoleptic properties of chicken patties. The results revealed that moisture content was significantly increased during the storage; the maximum moisture was observed in T0 (67.25% $\pm$ 0.03) on day 0, while the minimun was found in T3 (64.69% $\pm$ 0.04) on day 30. Supplemenatation of PUFAs in chicken patties significantly enhanced the fat content of the product the highest fat content was observed for T3 (9.7% $\pm$ 0.06. An increase in PUFAs concentration led to a significant increase in thiobarbituric acid reactive substances (TBARS). TBARS were increased from $1.22 \pm 0.43$ at 0 days to $1.48 \pm 0.39$ at 30 days of storage. The PUFAs incorporation negatively effected sensory acceptance of the product ranging from ($8.41 \pm 0.17$ to $7.28 \pm 0.12$). However, the sensory scores were in acceptable range for supplemented patties as compared to control sample. Treatment T3 depicted the highest nutritional content. The sensory and physiochemical analysis of supplemented patties suggested that PUFAs extracted from microalgae can be used as a functional ingredient in the preparation various meat products particularly chicken meta patties. However, antioxidants should be added to to prevent lipid oxidation in the product.

## INTRODUCTION

Consumer needs relating to the food production zone have changed in the past few years. Now a days, foods are preferred not only to satisfy hunger and source of nutrient provision for humans but also to treat nutritional conditions and enhance both physical and mental health (*Akram et al., 2022*). Fat is one of the three types of macronutrients responsible for providing energy. Chemically speaking, dietary fats may be broken down into three categories: saturated fatty acids, monounsaturated fatty acids, and polyunsaturated fatty acids (PUFAs) (*Saini et al., 2022*). Fatty fish like salmon and fish oils are considered the only good sources of PUFAs. There is an urgent need to provide alternatives to foods high in omega-3 fatty acids because most people in Western countries consume relatively little fish (*Chinarak et al., 2022*). Eggs and chicken meat are among the foods most commonly incorporated in meals for nutrition enhancement worldwide. For instance, in the United Kingdom, PUFAs are consumed primarily from chicken meat (73%) (*Dal Bosco et al., 2022*).

Saturated fatty acids are the most important contributor to heart disease among all other types of dietary fat (*Kyselová, Vítová & Řezanka, 2022*). A nationwide restriction on the consumption of saturated fats has been implemented. However, there has been no significant decrease in the incidence of heart disease over time (*Jadhav & Annapure, 2021*). The Mediterranean diet, which is quite popular, contains many unsaturated fatty acids, typically called "good" fatty acids. PUFAs are chains of hydrocarbons with two or more double bonds. Depending on where the first double bond is positioned, these fatty acids can be classified as omega-3, omega-6, or omega-9 (*Glaser, Heinrich & Koletzko, 2010*). The two simplest fatty acids (FAs) members of the omega-6 and omega-3 families are linoleic acid, also referred to as 18:2n-6 linoleic acid, and alpha-linolenic acid, also recognized as 18:3 omega-3 linoleic acid (*Calder & Yaqoob, 2009*).

The long-chain PUFAs with 20 carbons, such as dihomo-$\gamma$-linolenic acid (DGLA), arachidonic acid (AA), and eicosapentaenoic acid (EPA), are precursors for the biologically active eicosanoids known as prostaglandins, leukotrienes, and thromboxane. The long chain PUFAs with 22 carbons, such as docosahexaenoic acid (DHA), are precursors of decosanoids known as resolvins and protectins (*Calder, 2010*). In addition to these functions, PUFAs play a crucial part in the formation and function of cell membranes (*Calder, 2015*), as well as in the proper operation of the nervous system (*Singh, 2005*). Due to their participation in the inflammatory process and the impacts of eicosanoids and decosanoids, PUFAs are also important for normal functioning of immune system (*Hadley et al., 2016*). Moreover, proper fetus development depends on the presence of PUFAs in the mother's diet throughout pregnancy as AA and DHA omega-3 are considered essential for the growth of retina and nervous system in the fetus (*Molloy et al., 2012*). The literature reported by *Fukaya et al. (2007)* demonstrated that the antioxidant activity of PUFAs not

only imparts a protective effect against oxidative stress but also plays a role in preserving and functioning of the hippocampus.

Meat patties are a popular meat product across the globe among all age groups because of their low price, high nutritional value, convenient use, and sensorial satisfaction (*Mizi et al., 2019*). Generally, patties are made using beef or mutton however, chicken meat preferred due to its high-value proteins, mineral content, and lower lipid content (*Longato et al., 2017*). Meat companies traditionally use synthetic compounds to prevent oxidative deterioration and microbial spoilage, prolonging meat products' shelf life. Current studies have focused on utilizing natural compounds, particularly plant-based bioactive compounds, as an alternative to harmful synthetic antioxidants for meat products (*Passos et al., 2022*). Various natural extracts and functional ingredients have been used in meat products, but little research has been performed on supplementing PUFAs isolated from microalgae in chicken patties.

Previous studies revealed that the levels of DHA and EPA in the thigh meat increased to 2.4 and 1.3% of the total fatty acids when supplemented with fish oil at a dosage of 4% (*Konieczka, Czauderna & Smulikowska, 2017*). Another research found that adding 8.2% fish oil to chicken diets increased levels of all polyunsaturated fatty acids in the breast, particularly EPA (6.4%), DPA (3.1%), and DHA (6.4%). (7.8%) (*Güz et al., 2019*). Considering the importance of PUFAs and chicken meat in our daily diet, the current study was designed to develop PUFAs (omega-3) enriched chicken patties and to evaluate their physicochemical and sensory characteristics to predict the storage stability of the developed product.

## MATERIALS AND METHODS

### Raw material procurement

Raw materials (meat and dry ingredients) were procured from the local supermarket of Lahore and fetched to the functional food lab, department of food sciences, University of the Punjab, Lahore, Pakistan. The chicken meat was packed, labelled, and stored at $-18\,°C$, whereas all other dry ingredients were stored at room temperature. Extracted PUFAs were obtained from the biotechnology laboratory, National University of Medical Sciences (NUMS), Rawalpindi, Pakistan. All the chemicals used in the research experiment were of analytical grade obtained from ISI Pakistan Ltd.

### Extraction of PUFA from microalgae

Polyunsaturated fatty acids (PUFA) from algae were determined based on the methodology described by *Ikaran et al. (2015)* with slight modifications. Initially, an appropriate culture volume containing 30–40 mg of dry biomass was subjected to centrifugation ($5,500 \times g$, 10 min) and subsequently washed with deionized water to eliminate salts from the medium. The biomass was then subjected to freeze-drying. The extraction process was initiated by adding 0.7 mL of a mixture of chloroform: methanol (2:1) to the cell pellet, followed by 24 h of incubation at 50 °C with agitation. After centrifugation, the supernatant containing PUFA was transferred into a clean tube. The residue was re-extracted under identical conditions with 0.3 mL of chloroform: methanol (2:1), and the process was repeated until

**Table 1** Composition of chicken meat patties supplemented with PUFAs extracted from microalage.

| Ingredients | Treatments | | | |
|---|---|---|---|---|
| | T0 | T1 | T2 | T3 |
| Minced Chicken Meat (g) | 48 | 47.4 | 46.8 | 46.2 |
| Breadcrumbs (g) | 10 | 10 | 10 | 10 |
| Seasoning (g) | 1 | 1 | 1 | 1 |
| Ginger & Garlic (g) | 1 | 1 | 1 | 1 |
| PUFAs Oil (ml) | 0 | 0.6 (1%) | 1.2 (2%) | 1.8 (3%) |

a whitish pellet was obtained. The supernatants obtained from successive extractions were combined, washed with 0.88% KCl solution, and centrifuged for phase separation. The organic phase was transferred into a pre-weighed vial and evaporated using a rotational vacuum concentrator (RCV 2-18 CD, Christ, Germany) until a constant weight was achieved.

## Supplementation of PUFAs for development of chicken patties

Chicken patties were prepared according to the treatment plan elucidated in Table 1. Skinned chicken was properly ground using a meat grinder (8-mm diameter die plate) (MCR08 3.0, Arbel, Brazil) and combined with seasonings and PUFAs, whereas control was prepared without PUFAs. Patties were shaped by hand, and each patty was approximately 1 cm thick, weighed 80 g and was packed in LDPE bags. The product was stored at freezing temperature ($-18\,^\circ$C) analyzed on days 0, 10, 20 and 30 to evaluate compositional and physiochemical changes. Fat retention, cooking loss, and sensory evaluation were performed on fried patties, whereas proximate composition, TBARS, and microbiological assessment was conducted on raw chicken patties.

## Physicochemical properties of PUFAs supplemented chicken patties
### Nutritional profile of PUFAs supplemented chicken patties

The proximate composition of the chicken patties was analyzed using the standard methods of AOAC (2005). Mashed samples of chicken patties were oven dried at 110 $^\circ$C for 2 h, cooled in a desiccator, and moisture content was calculated as weight loss. Fat was extracted applying solvent extraction, ethanol was used as a solvent. The Kjeldahl method was used to assess crude protein in samples. Samples were subjected to muffle furnace at a temperature of 550 $^\circ$C for ashing, and the percentage of ash was calculated. Defatted samples were digested with acid and base to determine the amount of crude fibre. The remaining material was weighed and ashed after digestion. The difference in sample weight was used to compute the crude fibre.

## Determination of cooking loss of PUFAs supplemented chicken patties

The product was shallow fried in a De'Longhi fryer (Treviso, Italy) with canola oil at a temperature of 155 $^\circ$C for 25 mins. The product was weighed before and after frying, and cooking loss was calculated as weight difference according to the equation used by *Akwetey*

& Knipe (2012).

$$Cooking\ loss\ (\%) = \frac{final\ weight - initial\ weight}{Initial\ weight} \times 100.$$

## Determination of fat absorption in PUFAs supplemented chicken patties

Fat absorption was calculated using the methodology described by *Serdaroglu (2006)*, with modifications. Samples were weighed before and after frying and the given formulae was used to calculate the amount of fat absorbed by each sample. All the samples were run in triplicates, and results were recorded in percentage (%).

Fat absorption = % fat in fried meat patties − % fat in raw meat patties.

## Determination of pH of PUFAs supplemented meat patties

pH was measured using the method of *Soltanizadeh & Ghiasi-Esfahani (2015)*. 45 g of distilled water was used to homogenize the meat patty samples (5 g). The pH meter (Jenway, London, UK) recorded the analysis results.

## Assessment of TBARS in PUFAs supplemented meat patties

TBARS were assessed using the method of *Rahman et al. (2021)* with slight modifications. 50 ml of 20% trichloroacetic acid (TCA) was used to blend with 20 g meat patty sample for 2 min. A total of 50 ml water was used to rinse the blender content and filtered through a Whatman # 1 filter paper. After carefully combining 0.01 M, 5 ml of 2-thiobarbituric acid with the 5 ml TCA aliquot extract, the mixture was heated at 100 °C for an hour. The absorbance of the brown-orange color solution was determined in a spectrophotometer at a 532 nm wavelength. All samples were run in triplicates, and the amount of malonaldehyde in each sample was recorded as mg of malonaldehyde/kg.

## Microbiological evaluation of PUFAs supplemented meat patties

The total plate count technique was used to evaluate the microbial load of all samples. For each treatment upto four serial dilutions were made, and nutrient agar was used as growth media. The streak plate method was used to inoculate the plates with samples and incubate at 37 °C for 48 h (*Prommachart et al., 2020*). After incubation, colonies were counted, and results were recorded as CFU $\times$ $10^4$/g. The yeast and mold count of patties was measured using malt extract agar (MEA), plates were inoculated with diluted samples, incubated at 25 °C for five days with a constant supply of oxygen and results were recorded (*Manea et al., 2017*).

## Sensory evaluation of PUFAs supplemented meat patties

The nine-point hedonic scale was used to record sensory scores for organoleptic attributes of PUFAs supplemented meat patties. The product was served hot to 20 untrained and 10 trained panelists; sensory was performed in closed chambers with suitable sensory conditions such as proper light, no odors and water for rinsing. The product was

evaluated against color, odor, taste, and overall acceptability; scores were recorded and statistically analyzed as described by *Antonini et al. (2020)* to evaluate the effect of PUFAs supplementation on sensory attributes of meat patties.

## Statistical analysis

Descriptive statistics (mean, standard deviation) and inferential statistics ($t$-test, ANOVA) were used to analyze the primary objective of designed research. Data were examined using the statistical analysis and comparison programme SPSS version 25.0. For multiple comparisons, two-way ANOVA and LSD's post hoc analysis were performed. Outcomes were considered statistically significant, showing a $p$-value higher than 0.05.

## RESULTS

### Proximate composition of PUFAs supplemented chicken patties

The results observed for the proximate composition of the control sample and PUFAs enriched meat patties during one month of storage are elucidated in Table 2. Moisture content varied significantly ($p < 0.05$) by adding PUFA to the chicken. Moisture content increased dramatically during the storage interval but decreased with the addition of PUFA. It is evident from Table 1 that the highest moisture content was obtained for T0 at 30 days of storage ($68.07 \pm 0.04\%$), while the lowest moisture content was obtained for T3 at 0 days of storage ($64.16 \pm 0.08\%$). Regarding overall moisture content among treatments, the highest moisture content was obtained for T0, while the lowest moisture content was obtained for T3. In the case of overall moisture content during storage interval, the highest moisture content was obtained at 30 days of storage while the lowest overall moisture content was received at 0 days of storage. Protein content also decreased significantly ($p < 0.05$) with the increasing concentration of PUFA and storage interval. The highest protein content was obtained for T0 at 0 days of storage which was $16.94 \pm 0.06\%$, while the lowest protein content was observed for T3 at 30 days of storage at $16.01 \pm 0.01\%$. Fat content significantly increased with the addition of PUFA and storage days in the final product. Maximum crude fat was observed for T3 ($26.62 \pm 0.08\%$), while minimum content was obtained for T0 ($26.26 \pm 0.12\%$). At the same time, the content during storage interval showed the highest values for the product at 0 days of storage ($26.26 \pm 0.12\%$), while the lowest overall Fat content was obtained at 30 days of storage ($26.34 \pm 0.12\%$). The mineral content evaluation revealed a non-significant ($p > 0.05$) effect of PUFA and storage days on the meat patties. It was similar for all treatments, and no significant changes occurred in the ash content during 30 days of storage.

### Cooking characteristics

All the samples of patties were assessed for their cooking loss among treatments and during storage intervals. The results of the cooking loss in control and microalgae extracted PUFA samples are shown in Table 3. Results showed a non-significant ($p > 0.05$) variation in cooking loss of chicken patties supplemented with microalgae-extracted PUFAs. The maximum cooking loss was observed for the T3 sample, $30.52 \pm 0.44\%$, while the lowest was observed in the control sample, T0, $29.14 \pm 0.64\%$. Similarly, during the storage time

**Table 2  Proximate composition of chicken patties supplemented with PUFA extracted from microalgae.**

| Proximate composition (%) | | PUFA enriched chicken patties | | | |
|---|---|---|---|---|---|
| | | T0 | T1 | T2 | T3 |
| **Moisture** | 0 day | $67.25^d \pm 0.03$ | $66.29^h \pm 0.04$ | $64.82^l \pm 0.05$ | $64.16^p \pm 0.08$ |
| | 10 days | $67.52^c \pm 0.04$ | $66.50^g \pm 0.08$ | $65.01^k \pm 0.16$ | $64.30^o \pm 0.01$ |
| | 20 days | $67.63^b \pm 0.11$ | $66.80^f \pm 0.23$ | $65.29^j \pm 0.02$ | $64.40^n \pm 0.06$ |
| | 30 days | $68.07^a \pm 0.04$ | $67.14^e \pm 0.01$ | $66.02^i \pm 0.25$ | $64.69^m \pm 0.04$ |
| **Protein** | 0 day | $16.94^a \pm 0.06$ | $16.60^e \pm 0.03$ | $16.19^i \pm 0.01$ | $16.12^k \pm 0.02$ |
| | 10 days | $16.86^b \pm 0.04$ | $16.50^f \pm 0.01$ | $16.17^i \pm 0.01$ | $16.09^l \pm 0.01$ |
| | 20 days | $16.72^c \pm 0.11$ | $16.37^g \pm 0.02$ | $16.15^j \pm 0.01$ | $16.06^m \pm 0.01$ |
| | 30 days | $16.64^d \pm 0.01$ | $16.31^h \pm 0.05$ | $16.14^{jk} \pm 0.01$ | $16.01^n \pm 0.01$ |
| **Fat** | 0 day | $26.26^l \pm 0.12$ | $26.28^k \pm 0.16$ | $26.32^j \pm 0.09$ | $26.62^a \pm 0.08$ |
| | 10 days | $26.37^g \pm 0.12$ | $26.49^c \pm 0.16$ | $26.41^e \pm 0.31$ | $26.39^f \pm 0.38$ |
| | 20 days | $26.57b \pm 0.11$ | $26.36^h \pm 0.07$ | $26.37^g \pm 0.17$ | $26.14^m \pm 0.08$ |
| | 30 days | $26.34^i \pm 0.12$ | $26.41^e \pm 0.33$ | $26.48^d \pm 0.23$ | $26.41^e \pm 0.34$ |
| **Ash** | 0 day | $2.15^c \pm 0.12$ | $2.16^b \pm 0.16$ | $2.15^c \pm 0.09$ | $2.16^b \pm 0.08$ |
| | 10 days | $2.15^c \pm 0.12$ | $2.15^c \pm 0.16$ | $2.17^{bc} \pm 0.31$ | $2.18^a \pm 0.38$ |
| | 20 days | $2.17^{ab} \pm 0.11$ | $2.16^b \pm 0.07$ | $2.16^b \pm 0.17$ | $2.18^a \pm 0.08$ |
| | 30 days | $2.15^c \pm 0.12$ | $2.14^c \pm 0.33$ | $2.15^c \pm 0.23$ | $2.16^b \pm 0.34$ |

**Notes.**

Mean values ±S.D with different letter in individual parameters are significantly different ($P < 0.05$).

T0 (control), without PUFA; T1, chicken patties containing 1% PUFA; T2, chicken patties containing 2% PUFA; T3, chicken patties containing 3% PUFA.

of one month, the highest cooking loss was noticed at 30th day of storage ($29.56 \pm 0.35\%$), whereas the lowest cooking loss of ($29.15\% \pm 0.64$) was seen at the start of the study.

## Fat absorption

All the samples of patties were analyzed for their fat absorption among treatments and during storage interval. The results observed for fat absorption analysis of samples supplemented with microalgae-extracted PUFA samples during one month of storage are provided in Table 3. Results depicted a significant variation in the fat absorption values of meat patties. Fat absorption increased significantly ($p < 0.05$) with the concentration of PUFA and decreased during the storage interval at freezing temperature. In the overall fat absorption among all treatments, the highest fat absorption was observed for T3 ($5.87 \pm 0.03\%$), while the lowest fat absorption was obtained for treatment T0 ($4.90 \pm 0.02\%$). In the case of overall fat absorption during the storage interval, the highest fat absorption ($4.90 \pm 0.02\%$) was obtained at 30 days of storage. The lowest overall fat absorption was received at 0 day of storage ($4.77 \pm 0.02\%$).

## pH of microalgae extracted PUFA supplemented chicken patties

The pH of the chicken patties samples supplemented with PUFA during 30 days of storage is provided in Table 4. A significant decrease in pH value was observed ($p < 0.05$) with the rising concentration of polyunsaturated fatty acids in meat patties and with the increasing

**Table 3 Cooking characteristics of chicken patties supplemented with PUFA extracted from microalgae.**

| Cooking characteristics (%) | | PUFA enriched chicken patties | | | |
|---|---|---|---|---|---|
| | | T0 | T1 | T2 | T3 |
| Cooking loss | 0 day | $29.15^k \pm 0.64$ | $29.10^l \pm 0.01$ | $30.34^f \pm 0.01$ | $30.52^e \pm 0.44$ |
| | 10 days | $29.50^i \pm 0.23$ | $29.11^l \pm 0.01$ | $30.66^d \pm 0.07$ | $30.15^g \pm 0.44$ |
| | 20 days | $29.54^h \pm 0.44$ | $29.18^j \pm 0.04$ | $30.75^c \pm 0.01$ | $30.14^g \pm 0.64$ |
| | 30 days | $29.56^h \pm 0.35$ | $29.55^h \pm 0.01$ | $31.75^a \pm 0.06$ | $31.12^b \pm 0.23$ |
| Fat absorption | 0 day | $4.90^l \pm 0.02$ | $5.25^h \pm 0.04$ | $5.57^e \pm 0.15$ | $5.87^a \pm 0.03$ |
| | 10 days | $4.87^m \pm 0.01$ | $5.19^i \pm 0.01$ | $5.42^f \pm 0.03$ | $5.82^b \pm 0.01$ |
| | 20 days | $4.82^n \pm 0.04$ | $5.12^j \pm 0.05$ | $5.29^g \pm 0.01$ | $5.80^c \pm 0.01$ |
| | 30 days | $4.77^o \pm 0.02$ | $4.98^k \pm 0.09$ | $5.28^g \pm 0.01$ | $5.72^d \pm 0.07$ |

Notes.
Mean values $\pm$ S.D with different letter in individual parameters are significantly different ($P < 0.05$).
T0 (control), without PUFA; T1, chicken patties containing 1% PUFA; T2, chicken patties containing 2% PUFA; T3, chicken patties containing 3% PUFA.

**Table 4 Physiochemical characteristics of chicken patties supplemented with PUFA extracted from microalgae.**

| Physiochemical characteristics | | PUFA enriched chicken patties | | | |
|---|---|---|---|---|---|
| | | T0 | T1 | T2 | T3 |
| pH | 0 day | $5.94^a \pm 0.01$ | $5.78^b \pm 0.39$ | $5.56^d \pm 0.06$ | $5.14^k \pm 0.16$ |
| | 10 days | $5.64^c \pm 0.19$ | $5.56^d \pm 0.13$ | $5.50^e \pm 0.09$ | $5.40^h \pm 0.39$ |
| | 20 days | $5.55^d \pm 0.02$ | $5.51^e \pm 0.38$ | $5.51^e \pm 0.31$ | $5.50^e \pm 0.05$ |
| | 30 days | $5.45^f \pm 0.3$ | $5.42^g \pm 0.10$ | $5.38^i \pm 0.06$ | $5.33^j \pm 0.31$ |
| TBARS | 0 day | $0.65^p \pm 0.02$ | $1.10^l \pm 0.02$ | $1.45^h \pm 0.08$ | $1.71^d \pm 0.01$ |
| | 10 days | $0.83^o \pm 0.11$ | $1.15^k \pm 0.01$ | $1.50^g \pm 0.01$ | $1.82^c \pm 0.04$ |
| | 20 days | $0.96^n \pm 0.04$ | $1.22^j \pm 0.04$ | $1.58^f \pm 0.02$ | $1.88^b \pm 0.01$ |
| | 30 days | $1.02^m \pm 0.02$ | $1.29^i \pm 0.01$ | $1.62^e \pm 0.01$ | $1.90^a \pm 0.07$ |

Notes.
Mean values $\pm$ S.D with different letter in individual parameters are significantly different ($P < 0.05$).
T0 (control), without PUFA; T1, chicken patties containing 1% PUFA; T2, chicken patties containing 2% PUFA; T3, chicken patties containing 3% PUFA.

storage time at $-18$ °C. It was observed maximum for the control sample with no PUFA, which was $5.94 \pm 0.01$ and minimum for the sample with the highest concentration of PUFA, which was $5.14 \pm 0.16$.

## Thiobarbituric acid reactive species (TBARS)

The results observed for TBARS analysis of meat patties supplemented with PUFA during the specified storage time are provided in Table 4. Significant variations were observed in the TBARS data of samples during storage. The addition of polyunsaturated fatty acids (PUFA) significantly impacted the meat patties. Their TBARS increased from $0.65 \pm 0.02$ to $1.71 \pm 0.01$ for the control sample and T3 with the highest PUFA concentration, respectively, for fresh samples. While during storage time maximum value was observed

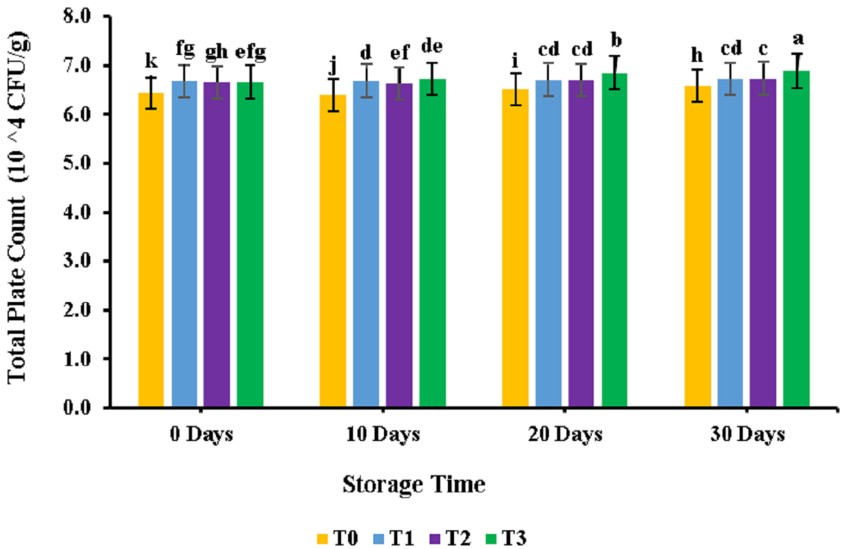

**Figure 1** **Total plate count ($10^4$ CFU/g) of chicken patties supplemented with PUFAs extracted from microalage.** Storage study with respect to TPC.

for the T3 sample on day 30, which was $1.90 \pm 0.07$ and the minimum was observed on day 0, given as $1.02 \pm 0.02$.

## Microbiological evaluation

The results for the total plate count analysis for meat patty samples are provided in Fig. 1. PUFA displayed a non-significant ($p > 0.05$) effect on the meat patties; similar results were observed for the control and PUFA-supplemented samples. However, the bacterial count increased significantly during the 30 days of storage. For the control sample, results revealed a bacterial count of $6.94 \pm 0.96$ CFU $\times 10^4$/g, while for sample T3, it was observed at $6.70 \pm 0.29$CFU $\times 10^4$/g. During storage, the bacterial number increased from $6.44 \pm 0.96$CFU $\times 10^4$/g to $6.58 \pm 0.09$CFU $\times 10^4$/g for the control sample. Similar trends were observed for samples treated with polyunsaturated fatty acids extracted from microalgae. Overall maximum plate count values were found for the T3 sample at 30 days of storage and minimum for the control sample at 0 days of storage.

All the samples of patties were analyzed for their yeast and mold among treatments and during storage interval. The results of the analysis are shown in Fig. 2. Analysis of variance revealed a significant ($p < 0.05$) effect of storage time on meat patties, while a non-significant effect ($p > 0.05$) of PUFA was observed. It can be seen from the figure that the highest yeast and mold was obtained for T3 at 30 days of storage. At the same time, the lowest yeast and mold were obtained for T0 at 0 days of storage. Similar results were observed for samples supplemented with PUFA and control samples (T0). Yeast and mold count found at 0 days of storage was $0.71 \pm 0.01$, while it increased to $0.86 \pm 0.08$ on the 30th day of storage for the control sample.
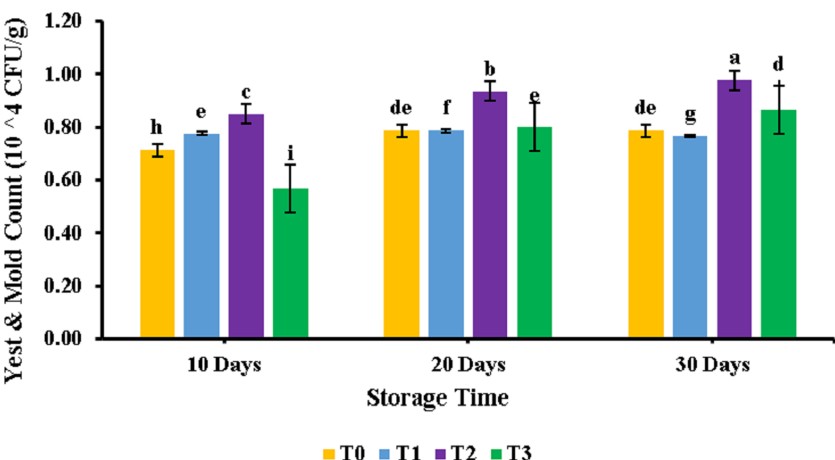

**Figure 2 Yeast and mold count of chicken patties supplemented with PUFAs extracted from microalage.** Storage study against yeast and mold.

### Sensory evaluation

All samples were analyzed for color, aroma, texture, taste, and overall acceptability. All the scores observed for samples are provided in Table 5. All the parameters were significantly affected by the supplementation of PUFA and storage time of 30 days. The color score of samples was reduced slightly by adding PUFA in meat samples and was found to be the minimum for the T3 sample at 30 days of storage. A similar trend was observed for the taste of samples supplemented with microalgae-extracted PUFA samples. Odor scores were observed to be reduced from 8.6 for the control sample to 7.5 for T3 on 0 days of storage. This score reduced from 8.35 to 7.26 on the 30th day of storage for the control and sample with maximum PUFA concentration, respectively. The texture of the samples was also significantly affected by PUFA addition and was observed maximum for the control sample at 0 days of storage. It was observed at 7.66 for the T3 and 8.56 for the control samples. In the overall acceptability scores among treatments, the highest scores were obtained for treatment T0, while the lowest were obtained for T3. In the overall acceptability score during storage interval, the highest was obtained at 0 days of storage while the lowest was obtained at 30 days of storage.

## DISCUSSION

During storage, increased moisture content of patties could be associated with released intracellular moisture and moisture gained from environment, similar findings were reported by *Dias et al. (2022)*, who incorporated canola and olive oil gel emulsions as pork fat replacers in beef patties. However, the decreased moisture content among the treatments could be interlinked to enhanced PUFAs, analogous results were reported by *Bavitha et al. (2016)* in fish burgers. *Serdaroğlu et al. (2018)* also reported increased moisture content during storage in beef patties formulated with dried pumpkin pulp and seed. The decrease in protein content during storage could be because there was an increase in moisture content

**Table 5 Sensory evaluation (scoring) of chicken patties supplemented with PUFA extracted from microalgae.**

| Organoleptic attributes | | PUFA enriched chicken patties | | | |
|---|---|---|---|---|---|
| | | T0 | T1 | T2 | T3 |
| **Color** | 0 day | $8.68^a \pm 0.02$ | $8.11^e \pm 0.01$ | $7.73^i \pm 0.04$ | $7.48^m \pm 0.06$ |
| | 10 days | $8.41^b \pm 0.01$ | $8.20^f \pm 0.01$ | $7.68^j \pm 0.04$ | $7.33^n \pm 0.03$ |
| | 20 days | $8.28^c \pm 0.01$ | $7.92^g \pm 0.07$ | $7.58^k \pm 0.02$ | $7.29^o \pm 0.02$ |
| | 30 days | $8.21^d \pm 0.03$ | $7.82^h \pm 0.03$ | $7.54^l \pm 0.01$ | $7.21^p \pm 0.11$ |
| **Aroma** | 0 day | $8.60^a \pm 0.03$ | $8.28^e \pm 0.01$ | $7.82^i \pm 0.01$ | $7.56^l \pm 0.02$ |
| | 10 days | $8.53^b \pm 0.04$ | $8.22^f \pm 0.06$ | $7.51^m \pm 0.02$ | $7.51^m \pm 0.03$ |
| | 20 days | $8.47^c \pm 0.01$ | $7.97^g \pm 0.04$ | $7.72^j \pm 0.02$ | $7.39^n \pm 0.12$ |
| | 30 days | $8.35^d \pm 0.03$ | $7.86^h \pm 0.05$ | $7.70^k \pm 0.01$ | $7.26^o \pm 0.02$ |
| **Texture** | 0 day | $8.56^a \pm 0.01$ | $8.21^d \pm 0.07$ | $7.85^g \pm 0.01$ | $7.66^k \pm 0.01$ |
| | 10 days | $8.55^a \pm 0.02$ | $8.03^e \pm 0.03$ | $7.83^h \pm 0.01$ | $7.59^l \pm 0.04$ |
| | 20 days | $8.53^b \pm 0.03$ | $7.91^f \pm 0.01$ | $7.78^i \pm 0.04$ | $7.43^m \pm 0.06$ |
| | 30 days | $8.43^c \pm 0.01$ | $7.93^f \pm 0.02$ | $7.72^j \pm 0.01$ | $7.23^n \pm 0.17$ |
| **Taste** | 0 day | $8.52^a \pm 0.21$ | $7.91^e \pm 0.14$ | $7.69^i \pm 0.01$ | $7.22^m \pm 0.01$ |
| | 10 days | $8.31^b \pm 0.08$ | $7.86^f \pm 0.03$ | $7.64^j \pm 0.01$ | $7.21^n \pm 0.01$ |
| | 20 days | $8.18^c \pm 0.04$ | $7.82^g \pm 0.06$ | $7.51^k \pm 0.01$ | $7.18^o \pm 0.02$ |
| | 30 days | $8.13^d \pm 0.02$ | $7.72^h \pm 0.02$ | $7.35^l \pm 0.04$ | $7.15^p \pm 0.01$ |
| **Overall Acceptability** | 0 day | $8.59^a \pm 0.02$ | $8.11^e \pm 0.01$ | $7.85^i \pm 0.01$ | $7.41^m \pm 0.01$ |
| | 10 days | $8.53^b \pm 0.04$ | $8.06^f \pm 0.01$ | $7.74^j \pm 0.07$ | $7.34^n \pm 0.01$ |
| | 20 days | $8.42^c \pm 0.03$ | $7.99^g \pm 0.04$ | $7.64^k \pm 0.01$ | $7.24^o \pm 0.05$ |
| | 30 days | $8.16^d \pm 0.02$ | $7.92^h \pm 0.01$ | $7.58^l \pm 0.06$ | $7.12^p \pm 0.02$ |

**Notes.**

Mean values $\pm$ S.D with different letter in individual parameters are significantly different ($P < 0.05$).

T0 (control), without PUFA; T1, chicken patties containing 1% PUFA; T2, chicken patties containing 2% PUFA; T3, chicken patties containing 3% PUFA.

that may have been subjected to the overall reduction in protein content. The decrease in protein among the treatments could be because of an increased polyunsaturated fatty acid addition. Analogous results were reported by *Bavitha et al. (2016)* in fish burgers during storage. The decrease may be attributed to the leaching out of the water-soluble nitrogenous components during storage along with moisture. Similar results were also reported in these studies when results revealed no significant change in ash content of fish burger patties during storage. In another study by *Serdaroğlu et al. (2018)* it was reported that no effect on ash content during storage in beef patties formulated with dried pumpkin pulp and seed powder was observed. During storage fat content decreases as there was an increase in moisture content that may have subjected to overall decrease in fat content. The increase in fat among the treatments could be because of increase in polyunsaturated fatty acid addition among the treatments. Very similar results were reported in a study on fish burgers during storage. *Liang, Zu & Wang (2020)* also reported decreased fat content during storage in PUFA-enriched eggs due to increased oxidation and TBARS values.

An increase in cooking loss was also observed during the one-month meat patties' storage time could be attributed to increased product moisture content during storage, which is the

main component reduced during cooking procedures. *Cortinas et al. (2004)* also reported similar results to the current study in which increasing the PUFA content diet of broiler chicken did not affect the fatty acids content of breast meat while cooking and revealed a non-significant effect. As patties lose their freshness over time, there is an overall increase in fat absorption during storage. Contradictory results displayed reduced fat absorption reported by *Soltanizadeh & Ghiasi-Esfahani (2015)* when aloe vera was added to low-meat beef burgers as a functional ingredient objective of improving their nutritional value.

There is a decrease in pH value among the treatments because of increased polyunsaturated fatty acid addition. *Troegeler-Meynadier, Bret-Bennis & Enjalbert (2006)* also reported similar results in their study of the effect of pH and concentration of linoleic and linolenic acids on the extent and intermediates of ruminal bio hydrogenation *in-vitro*. The increase in TBARS among the treatments was due to an increase in polyunsaturated fatty acid addition among the treatments. *Akarpat, Turhan & Ustun (2008)* also reported that increase in TBARS during storage in beef patties. *de Sousa et al. (2020)* reported similar results: TBARS increased in beef burgers enriched with PUFAs during storage in the refrigerator.

Among treatments, a significant trend was observed for total plate count, so it may be said that adding polyunsaturated fatty acids did not affect the total plate count in chicken meat patties. However, it depicted a rise during 30 days of storage time. *Karabagias (2018)* also reported similar results in chopped lamb meat during storage when bacterial count showed a significant surge during storage time. The increase in yeast and mold with storage patties can be the loss of their freshness and the increase in microbial load that may have been subjected to an increase in yeast and mould. Among treatments no significant trend was observed for TPC, so it may be said that adding polyunsaturated fatty acids did not affect the yeast and mold in patties. Similar results were also reported by another study performed on chopped lamb meat during storage (*Ibrahim, Abou-Arab & Salem, 2010*).

The decrease in texture scores during storage, the reason with storage is that patties lose their freshness, which may have resulted in an overall reduction in texture scores. The decrease in texture scores among the treatments could be because of increased polyunsaturated fatty acid addition. *Novello et al. (2019)* reported that adding golden flaxseed and by-product in beef patties increased their sensorial acceptance as Golden flaxseed added increased omega-3 fatty acids. *Valenzuela-Melendres et al. (2018)* also reported that adding PUFA sources to beef patties increases sensory acceptance. The decrease in color scores during storage could be because storage patties lose their freshness which may have subjected to an overall reduction in color scores. Enhanced concentration of PUFAs is associated with negative sensory scores for patties' colour. The decrease in taste scores during storage could be because storage patties lose their freshness which may have subjected to an overall reduction in taste scores. The decline in taste scores among the treatments could be because of increased polyunsaturated fatty acid addition. *Valencia, Ansorena & Astiasarán (2006)*, reported similar results when omega-3 PUFAs were supplemented to create nutrition-enriched fermented sausage. *Kolanowski et al., 2007* also reported that their sensorial scores were reduced with fish oil powder as a PUFAs source in instant food products.

## CONCLUSION

Polyunsaturated fatty acids are mainly taken from fish and fish products. They have a long history of being good for health. Some people do not typically eat fish, so other products are being produced with the artificial addition of PUFA. The current study revealed that adding indigenous polyunsaturated fatty acids sources can raise the PUFA content in chicken products. According to the findings of the study, moisture content increased considerably ($p < 0.05$) during storage while decreasing significantly ($p < 0.05$) across the treatments. Under retention, the protein content fell considerably ($p < 0.05$), and non-significant results were found among treatments. Ash content remained unaffected ($P > 0.05$) during storage or among the treatments. Fat content significantly ($p < 0.05$) decreased among the days while significantly ($p < 0.05$) increased during storage. TBARS significantly ($p < 0.05$) increased during storage and among the treatments. Highly significant results were obtained for the overall acceptability scores of patties with PUFA supplementation and during the storage period. Significant results were obtained from their interaction of storage time and treatments. The current study concludes that PUFA extracted from microalgae could be used as a supplement for the patties.

### Funding
The authors received no funding for this work.

### Competing Interests
The authors declare there are no competing interests.

### Author Contributions
- Sidra Dr conceived and designed the experiments, authored or reviewed drafts of the article, and approved the final draft.
- Muhammad Muneeb Zaman performed the experiments, prepared figures and/or tables, and approved the final draft.
- Zunaira Farooq conceived and designed the experiments, performed the experiments, authored or reviewed drafts of the article, and approved the final draft.
- Amina Hafeez analyzed the data, authored or reviewed drafts of the article, and approved the final draft.
- Muhammad Wasim Sajid conceived and designed the experiments, analyzed the data, prepared figures and/or tables, and approved the final draft.
- Muhammad Rizwan Tariq conceived and designed the experiments, analyzed the data, prepared figures and/or tables, idea was developed by Dr. Rizwan to work on this particular field, and approved the final draft.
- Shinawar Waseem Ali conceived and designed the experiments, performed the experiments, authored or reviewed drafts of the article, and approved the final draft.
- Sajid Ali analyzed the data, authored or reviewed drafts of the article, and approved the final draft.

- Muhammad Shafiq performed the experiments, authored or reviewed drafts of the article, and approved the final draft.
- Madiha Iftikhar performed the experiments, authored or reviewed drafts of the article, and approved the final draft.
- Waseem Safdar conceived and designed the experiments, analyzed the data, prepared figures and/or tables, and approved the final draft.
- Umair Ali performed the experiments, prepared figures and/or tables, authored or reviewed drafts of the article, and approved the final draft.
- Maria Kanwal conceived and designed the experiments, prepared figures and/or tables, and approved the final draft.
- Zujaja Umer analyzed the data, prepared figures and/or tables, and approved the final draft.
- Zunaira Basharat performed the experiments, authored or reviewed drafts of the article, and approved the final draft.

### Data Availability

The raw data is available in the Supplementary File.

### Supplemental Information

Supplemental information for this article can be found online at http://dx.doi.org/10.7717/peerj.15355#supplemental-information.

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
