# Peer review of "Supplementation of PUFA extracted from microalgae for the development of chicken patties"

_PeerJ, doi:10.7717/peerj.15355_

## Round 0.1 · original submission · Major Revisions

Please authors, kindly consider all the points raised by the reviewers. You can see there is a wide range of observations despite one reviewer accepting your submitted version, which shows there is merit of the work but others deemed room for improvement.

In addition, do pay close attention to the following:

a) Introduction requires more detail, especially in justifying why this work is relevant. Provide clarity on the gap that this work aims to occupy.

b) The discussion, please go more in-depth on the debate on the various results, particularly on the ‘why? and how? ‘

I look forward to your revised manuscript.

Reviewer 1 ·

Basic reporting

The manuscript titled "Supplementation of PUFA extracted from MIcroalgae for the development of Chicken Patties" has been clearly prepared. Its written in technical correct text. The manuscript contains sufficient introduction and background to demonstrate how the work fits into the broader field of knowledge. Relevant literature is appropriately referenced.
In addition, the structure of the manuscript conform to an acceptable format of ‘standard sections’. Figures are relevant to the content of the article and appropriately described and labeled.

Experimental design

The research fills an identified knowledge gap. Also, the research question in regards to a level of PUFA supplementation,is well defined. Selected methods are described with sufficient detail & information to replicate.

Validity of the findings

This study I consider, provides a high impact and novelty in the field. The application of Microalgae PUFA in formulation of patties showed a new implementation of PUFA origin and leads to an excellent results. Data proved they are robust and statistically sound. Conclusions are well stated, linked to original research question & limited to supporting results.

Additional comments

I strongly support to publish the presented work.

Reviewer 2 ·

Basic reporting

This poorly written manuscript with a lot grammar issuers does difficult to read. The authors should take their time and improve the manuscript (from introduction, methods, results and discussion). Below are some comments to help improve the manuscript
The abstract lacks material and methods ….please add. Also the abstract needs to be straightforward, concise and understandable.
Line 21 please define LC-PUFAs
Line 29 what is TBARS? Please define
Line 46 polyunsaturated fatty acids (PUFAs)
Line 100 appropriate temperature???? Like what?
Line 100 and proper handling before??? Meaning?
Line 100 PUFA extract was taken from the Biotechnology Laboratory, National University of Medical Sciences (NUMS), Rawalpindi, Pakistan.????? The authors should state exactly how the extraction was done. Also citation is need here or the extraction was novel?
Line 106 grounded by what?
Line 104 and 114 poorly described please improve
Line 129 poorly described please improve
Line 134 should be “of Rahman et al. (2021)” not of Rahman et al., 2021. Please correct this throughout the manuscript
Line 166 please improve
Line 169 “The statistical findings” whew!
Why are the p values not italized?

Experimental design

fair

Validity of the findings

fair

Reviewer 3 ·

Basic reporting

The manuscript 'Supplementation of PUFA extracted from microalgae for the development of chicken patties' has been a very interesting and well-executed research work. It has a strong premise and relevant hypothesis. The research objectives are quite clear and well-executed using standard protocols. The results were presented as data sets in form of tables and figures, and sufficient discussion has been provided with literature references.

Experimental design

The experimental design of the research work has been well structured to ensure good results.

Validity of the findings

The results were presented as data sets in form of tables and figures, and sufficient discussion has been provided with literature references.

Additional comments

General comments:
• The manuscript needs to be spelling and grammar checked.
• The abbreviations need to be mentioned with the term is being used for the first time in the manuscript. Successively, the abbreviation can be used without the term. This needs to be corrected in several places (Eg. PUFA, FA etc.,).
• The units of temperatures, proximate compositions etc., need to be placed at all times without fail.
• Many sentences have upper case letters in the middle of them, which needs to be corrected throughout the manuscript.
Specific comments:
1. Abstract: It has been mentioned as “n-3” in some places and “omega-3” in others. There need to be consistency in addressing the compound.
2. Abstract: The units need to be mentioned for each analysis, like moisture content, total fat etc.,
3. Line 33: The word ‘polyunsaturated fatty acids’ needs to be changed to ‘PUFA’.
4. Line 76: Why is the country Brazil mentioned specifically?
5. Line 100: The storage temperature needs to be mentioned.
6. Line 134: The year in the citation needs to be in brackets.
7. Line 262: Can any reference be added to support the statement?
8. Line 293: The word ‘in vitro’ needs to be italicized.
9. Line 317: Check the word ‘polyunsaturated fatty acid’.

Reviewer 4 ·

Basic reporting

1. Language needs to be improved in many places; some sentences are written very colloquially, and some are too complicated to understand (lines 20-24; 42-43). Check punctuation and e.g. Celsius signs. In my opinion, the first use of an abbreviation should explain its meaning, the reader does not always need to know what some abbreviations mean.

2. I found an error on one citation, the publication given doesn't exist, can you explain it? (Line 124- no corresponding article in the bibliography)

3. Please check all references to literature, there are errors, e.g. in the form of a misspelled name

4. The structure of the article is correct, although the text does not clearly show what the experimental setup looks like, I suggest preparing a diagram and specifying in detail which determinations were made before and after thermal treatment.

Experimental design

1. Experiment design was prepared correctly, although I am not sure which studies were conducted on raw patties and which on fried ones.

2. If possible, please provide the exact composition of PUFA oil (percentage of individual fatty acids), it would be good to prepare and compare the profile of fatty acids in finished products, it would significantly improve the quality of this publication.

3. I advise you to describe in more detail the method of the basic chemical composition

Validity of the findings

This publication has potential, but there are a lot of errors in it, that needs to be corrected. Starting with the language, sentence construction, or punctuation, which sometimes makes it difficult to understand. I also noticed some omissions in the results table. I am not sure about the veracity of the data. Small differences between repetitions are surprising. The description of the methodology should be improved, the methods used to determine the share of protein and fat were not described.

Additional comments

I suggest a major revision of this publication. There are many deficiencies when it comes to the language itself, and even more in the case of raw data (too small differences - which is impossible for such products using the described methods). Please improve the language to make it more understandable, maybe ask a fluent speaker to introduce corrections, please describe the methods of testing the fat, protein, and ash content in the samples, and prepare the experimental scheme. Please recalculate the standard deviations, because those in table 2 and those resulting from the data in the raw data table are different.

---

## Round 0.2 · accepted · Accept

The reviewers are very satisfied with the revised manuscript.

I agree that the current revised manuscript addressed all concerns, and is now acceptable for publication.

Thank you authors for finding PeerJ as your journal of choice, and I look forward to your future scholarly contributions.

Congratulations.

Reviewer 2 ·

Basic reporting

The authors have improve the manuscript

Experimental design

Good

Validity of the findings

Well explained

Additional comments

please accept the manuscript in the present form

Reviewer 4 ·

Basic reporting

The article after the corrections looks much better, most of the suggestions that appeared after the first review have been taken into account, the abbreviations have been clarified, and the language has been significantly improved.

Experimental design

The experiment is planned correctly, the details of each of the analyzes are described. The sense and significance of this experiment for science was explained. The research problem was explained in detail.

Validity of the findings

As for the processing of the results, it has been improved, especially in terms of standard deviation. The conclusions are correct and the results have been discussed with other similar papers published a little earlier.